# Coordination and Management of COVID-19 in Africa through Health Operations and Technical Expertise Pillar: A Case Study from WHO AFRO One Year into Response

**DOI:** 10.3390/tropicalmed7080183

**Published:** 2022-08-15

**Authors:** Nsenga Ngoy, Ishata Nannie Conteh, Boniface Oyugi, Patrick Abok, Aminata Kobie, Peter Phori, Cephas Hamba, Nonso Ephraim Ejiofor, Kaizer Fitzwanga, John Appiah, Ama Edwin, Temidayo Fawole, Rashidatu Kamara, Landry Kabego Cihambanya, Tasiana Mzozo, Caroline Ryan, Fiona Braka, Zabulon Yoti, Francis Kasolo, Joseph C. Okeibunor, Abdou Salam Gueye

**Affiliations:** 1World Health Organisation, Regional Office for Africa, Emergency Preparedness and Response Programme, Cité du Djoué, Brazzaville P.O. Box 06, Congo; 2Centre for Health Services Studies (CHSS), University of Kent, George Allen Wing, Canterbury CT2 7NF, UK; 3World Health Organisation, Regional Office for Africa, Emergency Preparedness and Response Programme, Nairobi Hub, United Nations Office in Nairobi UN Avenue Gigiri, Nairobi 00100, Kenya

**Keywords:** coronavirus, coordination, health operations and technical expertise, AFRO

## Abstract

**Background:** following the importation of the first Coronavirus disease 2019 (COVID-19) case into Africa on 14 February 2020 in Egypt, the World Health Organisation (WHO) regional office for Africa (AFRO) activated a three-level incident management support team (IMST), with technical pillars, to coordinate planning, implementing, supervision, and monitoring of the situation and progress of implementation as well as response to the pandemic in the region. At WHO AFRO, one of the pillars was the health operations and technical expertise (HOTE) pillar with five sub-pillars: case management, infection prevention and control, risk communication and community engagement, laboratory, and emergency medical team (EMT). This paper documents the learnings (both positive and negative for consideration of change) from the activities of the HOTE pillar and recommends future actions for improving its coordination for future emergencies, especially for multi-country outbreaks or pandemic emergency responses. **Method:** we conducted a document review of the HOTE pillar coordination meetings’ minutes, reports, policy and strategy documents of the activities, and outcomes and feedback on updates on the HOTE pillar given at regular intervals to the Regional IMST. In addition, key informant interviews were conducted with 14 members of the HOTE sub pillar. **Key Learnings:** the pandemic response revealed that shared decision making, collaborative coordination, and planning have been significant in the COVID-19 response in Africa. The HOTE pillar’s response structure contributed to attaining the IMST objectives in the African region and translated to timely support for the WHO AFRO and the member states. However, while the coordination mechanism appeared robust, some challenges included duplication of coordination efforts, communication, documentation, and information management. **Recommendations:** we recommend streamlining the flow of information to better understand the challenges that countries face. There is a need to define the role and responsibilities of sub-pillar team members and provide new team members with information briefs to guide them on where and how to access internal information and work under the pillar. A unified documentation system is important and could help to strengthen intra-pillar collaboration and communication. Various indicators should be developed to constantly monitor the HOTE team’s deliverables, performance and its members.

## 1. Introduction

The first human case of Coronavirus disease 2019 (COVID-19), caused by the SARS-CoV-2 virus, was identified in Wuhan in the People’s Republic of China on 30 December 2019 [1]. Soon after, the World Health Organisation (WHO) declared the outbreak as a public health emergency of international concern (PHEIC) and as a pandemic on 30 January and 11 March 2020, respectively [2]. Following the first COVID-19 case importation into Egypt on 14 February 2020, the disease has spread to the whole continent. As of 25 February 2021, a total of 111,762,965 COVID-19 confirmed cases (distributed as per Figure 1) and 2,479,678 deaths had been recorded, with 2.5% (2,789,965) of the cumulative global cases and 2.9% (71,204) of cumulative global deaths coming from the WHO African Region.

The number of cases and deaths recorded in Africa when writing this paper (March 2021) was much lower than predicted [3]. Several reasons have been advanced for these low numbers, including the role of aridity and temperature in transmission, demographic characteristics (distribution of age), the difference in identification of cases, and death detection capacity [4,5,6,7,8,9], and the possible contribution of pre-existing immunity from other viral infections [10]. Others have indicated that the numbers are due to the underestimation of the true magnitude of the pandemic resulting from weak surveillance systems [11,12] as postulated by the low rate of testing per population in the continent with ratios of as low as 1072 or 1441 tests per one million population in South Sudan and Niger, respectively [13].

However, the extent to which each (or a combination or interaction) of these factors has impacted the relatively low number of cases and deaths is yet to be fully explored. Nevertheless, what is clear is that the vast experience of responding to frequent outbreaks and emergencies has put the African Region in a comparatively better prepared position, and hence could mobilise the response capacity better than other regions. Besides the existing challenges of poverty and fragile health systems, the COVID-19 pandemic contributed to the disruption in its socio-economic activities in Africa, such as the breakdown in the delivery of health services [14]. The disruptions were largely from the measures put forward to curb the spread of the COVID-19 and included lockdowns, closure of borders and schools, restriction of travel, trade, and mass gatherings. These actions (such as border closures and lockdowns) taken by the countries in Africa helped slow the spread of COVID-19 on the continent.

In addition, most countries in the region rapidly instituted the incident management support team (IMST), a WHO system for coordinating and managing public health events in line with the WHO Emergency Response Framework [15]. The IMST is based on recognised best practices of emergency management included within the health sector. The critical functions for emergency response under the IMST are leadership, partner coordination, information and planning, health operations and technical expertise (HOTE), operations support and logistics, and finance and administration [16]. Prior to the first reported COVID-19 case in the WHO Africa region, a preparedness IMST was activated to assess, prepare, monitor, detect, and rapidly respond to the first case. Due to the nature of the emergency (involving all 47 countries in the region), an inter-cluster IMST structure that included repurposed staff from across the different clusters in the AFRO regional office was activated. The IMST is headed by the regional director with a designated incident manager (IM) that deals with the daily operations of the response. The IMST meets daily to share information and discuss strategic and operational issues to guide each country’s pandemic support and follow up action points for the technical staff to act. Figure 2 shows the first IMST structures for the AFRO pandemic response, which was then revised in April 2020 to Figure 3 after a regional interaction review (IAR) was conducted, which was necessary due to the protracted nature of the pandemic (the revised structure is discussed in detail below).

This paper documents the learnings (both positive and negative for consideration of change) from the activities of the HOTE pillar and recommends future actions for improving its coordination for future emergencies, especially for multi-country outbreaks or pandemic emergency responses. The HOTE pillar comprises of five sub-pillars, namely case management (CM), infection prevention and control (IPC), laboratory support, risk communication and community engagements (RCCE), and emergency medical teams (EMTs), all of which involve specific technical expertise that focuses on building a country’s capacity and providing technical and operational support in the response [17,18]. We focus on the HOTE because it is the core operational and interventional pillar, forming the driving force of response under the IMST and because its operation exceeded the usual country-level cooperation that the WHO country office (WCO) has with the member states [16]. While other IMST pillars are important, the HOTE pillar plays a significant role in linking the WHO with the Ministry of Health (MOH) and its partners to ensure optimal coverage and quality of health services in response to emergencies by promoting the implementation of the most effective, context-specific public health interventions and clinical services by operational partners. For instance, the pillar links directly with the member states’ MOHs by providing SOPs, technical guidelines, the best practices and protocols that promote adequate responses and quality of health services in response to emergencies. Through the guidelines, HOTE fosters the implementation of the most effective and context-specific public health interventions. Adequately, the pillar assesses different interventions to provide a regional profile that guides the MOHs need to deploy experts on the subject matter and organise virtual and onsite training needs. HOTE also ensure optimal coverage and clinical services by linking with operational partners and providing essential supplies such as personal protective equipment, medical oxygen, etc. Further, unlike other pillars in the IMST, the focus on the HOTE pillar is to provide up-to-date, evidence-based field operations, policies, and guidance.

## 2. Methods

A mixed methods, case study methodology as defined by Yin [19] was utilized in this study. It focuses on the coordination and response to the COVID-19 pandemic from the HOTE pillar’s perspective. The parameters studied included all activities implemented by the HOTE pillar one-year post activation of the IMST for the pandemic in the AFRO region as defined in the comprehensive Strategic Preparedness and Response Plan (SPRP) February-December 2020 updated (May 2020) [20]. The analysis was done using mixed methods to converge the findings to increase validity and to have individual components complementing each other, thus providing better explanations for the phenomenon under investigation [21].

Firstly, we retrieved the retrospective information on the activities and operations of the HOTE pillar, background information, historical insight into the pillar work, and a picture of how WHO AFRO or its emergency program fared over time [22]. The information was retrieved through a document review of available reports and surveys, periodicals, and monthly bulletins accessed through the WHO website. Those that were not publicly available such as HOTE minutes, confidential reports, and manuals, were gathered from the key informants who were part of the HOTE team from the onset of the pandemic. Additionally, the pandemic response staff from the two regional hubs of WHO (Dakar and Nairobi) and the planning, monitoring, and evaluation cell were requested to provide any other relevant documents and information on local policies, strategies and work plans that were deemed useful for the study.

Secondly, key informant interviews (n = 14) were conducted with purposively selected staff heading different sub-pillars in HOTE and the focal persons supporting individual countries who had been involved since the pandemic began and since the pillar was instituted. One researcher (BO) conducted the interviews using a semi-structured interview guide that was developed based on the content and gaps identified from the document review. The interviews were used to verify findings or corroborate evidence from the document reviews [22]. All the participants who agreed to undertake the interviews were invited to participate after being explained for the study purpose, and they gave verbal informed consent. The interviews were conducted in English and audio-recorded, and each lasted between 30–45 min. All the IDIs were transcribed verbatim in English and compared against their respective audio files by one researcher BO. All the validated transcripts were extracted in MS Excel for ease of management and transparency of the analysis process. All information/data provided from the document review and key informant interviews were synthesised using thematic analysis into thematic areas around learnings.

## 3. Findings

The pandemic response revealed that shared decision making, collaborative coordination, and planning have been significant in the COVID-19 response in Africa. The HOTE pillar’s response structure contributed to attaining the IMST objectives in the African region and translated to timely support for the WHO AFRO and the member states.

### The Changes in the Organisational Structure of Health Operations and Technical Expertise Pillar during the Pandemic and the Roles of the Sub Pillars

The HOTE is a critical pillar in the COVID-19 response and ensures that optimal and quality guidance of emergency response services are effectively communicated to African region countries. This pillar also provides updated evidence-based field operations, policies, guidance, and technical expertise during the response. In January 2020, during the preparedness phase and the initial stage of the outbreak, the HOTE pillar had ten sub pillars: influenza, CM, IPC, laboratory support, RCCE, vaccines and immunisation, POE/support to the operation, research, service delivery/health services continuity, and capacity building. Following the global pandemic declaration in March 2020, the AFRO region IMST structure was revised with the HOTE pillar reduced to six sub pillars: influenza, vaccines and immunisation, POE/support to the operation, research, and capacity building. As the pandemic evolved, the HOTE pillar was further revised in August 2020 to five sub-pillars: CM, IPC, laboratory support, RCCE and emergency medical teams (EMT). Additionally, through the WHO AFRO emergency hubs in Dakar, Senegal and Nairobi, Kenya, the HOTE pillar working with the country focal persons provided/s daily strategic support to countries on the various operational and response activities. The country focal points (CFP) team was created when the size of the pandemic grew, requiring that the HOTE responds to a multiplicity of country requests. The CFPs were constituted to provide frequent engagement with the countries to monitor the evolution of the pandemic and flag issues of support. The criteria used for modification of the HOTE pillar was based on the findings of the Intra Action Review (IAR) carried out on the IMST, which provided lessons to situate better the pillars as part of the Incident Management System (IMS) as guided by the Emergency Response Framework (ERF) [16].

Currently, the HOTE pillar activities coordination is led by a pillar lead who coordinates the five sub pillars’ operations (Figure 3). The pillar lead participates in the cross-pillar meetings at the IMST strategic meeting and is the pillar’s voice to the management. Having a HOTE pillar team lead working with the sub pillar leads has been an effective way of outlining the pillar needs, making strategic guidance on the management of COVID-19 across the region, and identifying the gaps that need the attention of the administration. The pillar lead has also been able to link the team to other pillars and units (such as the CFPs) to update the regional management on the countries’ ongoing preparedness and response activities (Table 1). Overall, the operations of the HOTE pillar have focused on the dissemination of strategic and technical guidance adapted to regional contexts; reinforcement of capacities, including the deployment of experts; further expansion of laboratory diagnostic capacities for COVID-19 in all countries; resource mobilisation at regional and country levels; and supporting the distribution of essential supplies such as personal protective equipment (PPE), laboratory equipment and reagents, and other medical devices to member states. HOTE pillar support to countries at regional, national, and sub-national levels was clearly defined in a comprehensive SPRP February-December 2020 updated (May 2020) [20].

## 4. Key Learnings

### 4.1. HOTE Intra and Extra Team Coordination

The HOTE pillar holds a weekly strategic meeting where the sub-pillars present updates on key activities undertaken and plans for the upcoming week. In addition, challenges and issues experienced by individual countries are also presented and discussed to provide solutions. In collaboration with the cross-pillar lead and using available data, deep-dive discussions are held to address cross-cutting issues in countries and outcomes of these discussions are presented at the main IMST meetings. At the pandemic’s peak, cross pillar discussions were held every week, and were further relaxed to a biweekly basis as the country’s epidemiological situation improved. Concerning the regional IMST meetings, daily meetings were held with all pillar leads and team members (including senior management representations) at the start of the pandemic. As the regional epidemiological situation improved, the frequency reduced to three times a week and two times a week as from March 2021.

Given that daily IMST meetings were a useful platform for information sharing and learning, the HOTE pillar lead or a designated pillar team member regularly presented to the IMST on the pillars ongoing activities with other team members providing additional information comments.

The cross-pillar discussion helped provide solutions to the ongoing challenges that countries faced, e.g., the surge response to South Africa. It allowed the staff better to understand the impact of the new disease- COVID-19. As shown in Figure 3, the CFPs are part of the AFRO IMST for COVID-19 and provide the overall country coordination, guidance, technical and operation support to WHO country offices (WCO) regarding emergency response management under the IMST. They link the WCO and IMST at the regional office and headquarters. Functionally, they report to the HOTE team lead as per the IMST. They perform information management roles (such as ensuring WCO sitrep and the dashboard of major events in countries are updated and shared regularly) and monitor and follow up on the countries’ operational support. Otherwise they are involved in resource mobilisation in-country and externally, and follow up with the implementation of the response plan and utilisation of mobilised funds. Other roles include surge support for HR and supplies; planning, managing, and monitoring performance standards and key performance indicators during operations response; conducting the countries’ needs, gaps, and capacity assessments/analyses; and leading capacity building roles.

Therefore, the collaboration between the CFPs and the HOTE was important as they brought the various challenges that countries were facing, the role or support that the AFRO IMST provided to the member states, and the country’s best practices key upcoming events.

While the HOTE team members would also attend the country meeting to respond to questions and requests for the WHO AFRO management, the CFPs returned the key responsibility of leading discussions with countries. Equally significant was the lateral coordination and collaboration between HOTE and other pillars (such as epi-analytics (see Figure 3 for more sub pillars)). The pillars within the IMST are interlinked, and each contributed to the other’s work, with the overall outcome being the achievement of the strategic objectives under the COVID-19 Strategic Preparedness and Response Plan (SPRP) 2020 [17,18].

For instance, the collaboration between epi-analytics and HOTE teams would generate cross pillar presentations that summarise a country’s concerns or challenges, such as evidence of increasing healthcare workers infection or poor case management. This strategic information and solutions are subsequently shared with the AFR senior management for endorsement of proposed actions. As a result of the strategic role that the HOTE pillar played, it was noted that countries would get in touch directly with the HOTE team members to follow up on the envisaged/expected supports; thus demonstrating that countries appreciated the support they were getting from HOTE.

### 4.2. Internal and External Coordination Meetings

The internal and external coordination meeting of the HOTE pillar went through several changes as the pandemic evolved. At the start, sub-pillars under the HOTE pillar would meet during a weekly meeting at which the challenges and planned activities for the week would be discussed. From this sub-pillar collaboration, it was easy to learn what the other teams were doing, and it allowed pillars to request support help from others. Additionally, alternative discussions through communication by email normally brought about a significant response from different sub pillars whenever it was needed, as a result of the comradeship created through the weekly meetings. Furthermore, to enhance the coordination of various sub pillar components, the team worked closely with the emergency Hubs, whose roles were linked to the team. This coordination process effectively enabled policymakers to set forth actions that ensured best practices with the desired goal [23].

During the early phase of the pandemic, the general HOTE coordination meetings were poorly attended and unnecessarily long. The poor attendance may be attributed to the timing and conflicting meetings that the different pillar members attended, which subsequently led to the slow implementation and follow up of planned actions. Also, given that the various sub-pillar members were working virtually from different countries, a difference in the time zones may have been a factor in the poor attendance at the HOTE meetings. Thus, coordinating HOTE meetings was a major challenge and required cooperation from all involved to achieve the desired objective of the pillar.

### 4.3. HOTE Collaboration with Countries

Early in the pandemic, there was non-synchronised coordination between the HOTE pillar with the countries. The sub pillar team members would join in the different country’s meetings and teleconferences (TCs) to understand the countries’ challenges and, where possible, offer on-the-spot technical guidance. In some countries it yielded positive results, as some countries that reached out to HOTE pillar members received help with operational and technical needs requested.

With time, the HOTE meetings with member states focused on helping the countries meet their objectives/goals. Through this collaboration, countries developed trust in the support provided by the HOTE pillar team and would engage in unrelated areas surrounding their respective ministries of health. Requests whose solutions were not readily available were referred to the AFRO management support. A provision of the solution to the countries strengthened the WHO’s credibility in the countries and provided an opportunity for countries to improve their health workforce and health systems.

However, the attendance of countries TCs by HOTE sub pillar members and the designated country focal points based at the hubs were met with resistance and were perceived negatively in some countries. Moreover, the WHO AFRO through HOTE or other region pillars would be demurred by some countries as making demands for information rather than supporting and offering solutions to the response. Working through the CFPs and the two hubs as the entry point to the countries was a more acceptable way of coordinating the support of member states. However, the CFPs assigned to respective countries would be overwhelmed with the meetings and follow-ups, given that they were concerned with coordinating other activities other than just the pandemic.

On several occasions, countries preferred to directly interact with the HOTE team members who were thought to have additional capacity to address their requests. Occasionally, this was also due to the fact that urgent information requested by countries sent through the CFPs by the HOTE team was not transmitted to the respective countries on time, thus causing delays.

In other cases, when the countries dealt directly with the CFPs—especially when following up on financial support—the CFPs would reach out to the HOTE team members directly and follow up the same work. On some occasions, when the CFPs would be on leave or have conflicting activities from other assignments, there would be gaps in the countries’ meeting attendance and communication. There were also delays in sourcing information from teams in member states, which hampered much-needed interventions to curb the pandemic. The other pillars were equally affected, resulting in a resurgence of the cases in most countries and increased mortality. However, when the CFPs communication teams were more established and straightened, a collaborative approach was taken, which led to reduced delays.

### 4.4. Intra-Communication among HOTE Members

Clear, well-defined communication creates an effective team, although this is usually an insidious process [24]. At the onset of the pandemic, there was a lack of clarity about the different sub-pillar members’ responsibilities and roles and the communication channels between the different sub-pillars to management. This was particularly evident with some sub-pillars not creating time for orientation/briefing of new team members who joined at different times during the pandemic, making it difficult for the team to be on the same page. There appeared to be some team members who did not understand their roles and responsibilities from the onset, and this made it difficult for them to fit in the HOTE pillar structure.

To address the above, sub-pillar team leaders defined the roles and responsibilities for each member. The new team members had difficulty finding their way around the organisation. For instance, it took a long time to have the members set up with email systems to ease coordination within the response structure, and when it was done, it was not very clear whom/where to access information. Consequently, failing to get the new members timely access to the organisation’s email account hampered their coordination work with the countries, since some countries did not respond to HOTE members communicating with them through personal email accounts. Nonetheless, with some countries’ ways of working (not responding to emails from personal mail addresses without the WHO domains and feeling bombarded), it remains a challenge to get a timely response from them.

It was not always easy to get information about internal coordination mechanisms and meetings within WHO. When some members participated in the discussions, it appeared as though they didn’t understand what the meeting was all about and thus would not effectively implement session follow up actions. Because of the lack of introduction/induction of new HOTE and IMST members, not all team members within the sub pillars were receptive, as some could not respond to emails from the unknown (new) team members. The lack of adequate orientation of the members was most significant, especially for those who came mid-pandemic and did not know where to go or understand the organisation’s processes. In some instances where orientation was provided, there would be a lack of guidance on accessing the technical reports, which may have resulted in duplication of efforts in producing some reports and briefs. It was, however, easier to get information from the management and incident management leadership than at the technical level.

### 4.5. Documentation and Information Management

We find that multiple information sharing channels under the HOTE pillar were conveying different information about the pillar. Potentially this may have fuelled different results among the sub pillars, causing poor utilisation of the available information to guide the ongoing pandemic support. There was a significantly large amount of information produced and collected by the different sub pillars that was not being utilised for strategic decision making, or that would require a unified way of constellating. The team members’ strength was more geared towards providing technical guidance to the member states rather than synthesising the information. At the later pandemic stage, the emergency information management was set within the HOTE pillar to support information synthesis and consolidation between the different sub pillars to aid in strategic decision making and unified documentation. This helped strengthen the intra pillar information management. However, gathering information between some of the sub pillars was still not straightforward.

### 4.6. Monitoring Objectives

Based on the results, it was difficult to ascertain whether the HOTE pillar had achieved its intended objective. This was because the pillar did not have objectively verified indicators that they would use to track the pillar’s performance during the year; however, the sub-pillars had indicators. At the HOTE pillar’s operations level, it felt like the pillar was working without a strategy. When it came to writing the different reports for the pillar, each team was writing in an unfocused and uncoordinated way without really focusing on the indicators that the HOTE pillar wanted to achieve. Overall, this made it hard for HOTE to report achievement against indicators.

## 5. Discussion and Conclusions

The coordination and management of the COVID-19 pandemic is a complex undertaking that requires consolidated effort in synchronising actions and multilateral decisions. A well-coordinated response is the fastest way to transition out of a pandemic [25], and this entails skilled professionals being fully capacitated to respond to public health emergencies [26]. As previously shown by WHO [27], the early stages of an outbreak require various synergic components such as coordinating response, communication, ensuring health intervention, and managing information. Similarly, these components should have amenable objectives that will characterise the scope of activities [28]. The results under the HOTE pillar have shown that shared decision-making, collaborative coordination, and planning have been significant in the COVID-19 pandemic response from an African context.

The AFRO region IMST objectives are spelt out in the COVID-19 SPRP 2020 [17,18]. While the objectives of each of the sub pillars under the HOTE pillar’s response structure are included in the SPRP, the findings on monitoring and evaluation have shown that it is difficult to ascertain whether the HOTE pillar had achieved its intended objective. Interestingly, the amalgamation of some sub pillars to HOTE came mid-response, and the SPRP 2020 outlined the sub pillar objectives while the SPRP 2021 was created to improve the objectives. However, through the response structure, the sub-pillar teams have been able to timely support the member states (particularly allowing countries to operationalise the in-country response’s management structure as defined based on the country plans). Additionally, the structure has been useful in notifying the senior management at WHO AFRO on the needs (for the countries) and strategic directions.

However, the results have also shown that while coordination is critical in standardised situations, uncertainties and complex scenarios present challenges for coordination research practices [29]. For instance, the limitations on the coordination meetings’ attendance, intrapillar interaction challenges, and even the countries’ challenges all may have hampered the progress of the response from the onset. However, given the changes made progressively as the pandemic progressed, it was easy to support the countries. As observed in the management of the Ebola outbreak in West Africa, the separation of the technical and operational components of the response coordination under the IMST streamlined the coordination and enabled the experts to support their technical work [30]. The experience with the coordination of the HOTE pillar builds into the framework proposed by Hernantes et al. [31] and addresses the four specific challenges characterising crisis management, including the heterogeneity of the actors and stakeholders involved, multi-dimension effects, the diversity of activities to build resiliency, and the centrality of knowledge transfer and sharing mechanisms.

Nonetheless, we believe that sustaining the coordination approach could enhance IMST management of pandemic objectives of a similar magnitude in AFRO. We propose the following recommendations to strengthen the approach.

The main strength of this study was the triangulation of data/information from the document review and interviews, which provided a platform for gaining a deeper and broader understanding of the different coordination aspects of the HOTE pillar. The main limitation was capturing the learnings as both positive and negative aspects rather than using a framework. However, the analysis was adequately driven by data through thematic analysis.

## 6. Recommendations

As WHO AFRO continues to support the member states, we propose changes to the work approaches that could tackle the challenges that our review has revealed. For instance, we suggest that both internal and external meetings have agendas shared in advance and that the calendar invites are shared with all the members in advance. This would allow participants to prepare in advance and even adequately engage with the content of the meetings. Through this, it will also be possible to sufficiently capture the notes for the records of the meetings and share them with participants to track the meeting action points or requests from member states. Moreover, requested information could be streamlined to key questions from the countries, CFPs, and the two regional hubs. Once streamlined, there is a need to agree on how the information could flow through the country hubs or focal points and then to HOTE or even IMST. This could help define a unified flow of information and a better understanding of the countries’ challenges and reduce the relayed responses from them. Also, we propose that we have a streamlined way of requesting information and come together as a pillar to submit our work.

Besides, the roles and responsibilities of each pillar member should be streamlined and identified. This would enhance better knowledge and performance of each member and even ease the justification of the roles to the senior management. It would also be imperative to have a HOTE introductory note indicating where and how to access new members’ information, besides having an orientation to understand the organisations’ way of working. The notes and the orientation would help set up the new members joining the team and share the visions and objectives of the different sub pillars. It would help the team remain strategic in achieving its key goals and expectations without delays and making the members who join mid emergency feel welcome. Also, at the onset of the teams, it is imperative to have key performance indicators (KPIs) for each member to achieve a common goal. This is essential to help to improve our strength in our systems and improve the quality of care. Besides, continued collaboration between different pillars—given that we are intricately woven together—would produce better performance outputs.

There is a need to continue working towards achieving a unified documentation component so that the pillar can speak from a unified position. This helps to strengthen what we do as a pillar, and it also allows the pillar to communicate to IMST on the activities that we do, and they can contribute directly to IMST objectives. Besides, it provides a unified way to guide the member states and gives us an opportunity and the sub pillars to focus our energies on providing technical guidance to countries.

The sub pillar indicators should be defined to the team to know whether the intended performance is being achieved or not. Additionally, it would be imperative to add coordination level indicators to examine the role of coordination in achieving the sub pillars’ objectives. Constantly monitoring the deliverable indicators’ performance would ensure that we are—as a team—discussing the problems (technical and financial) and identifying the solutions.

## Figures and Tables

**Figure 1 tropicalmed-07-00183-f001:**
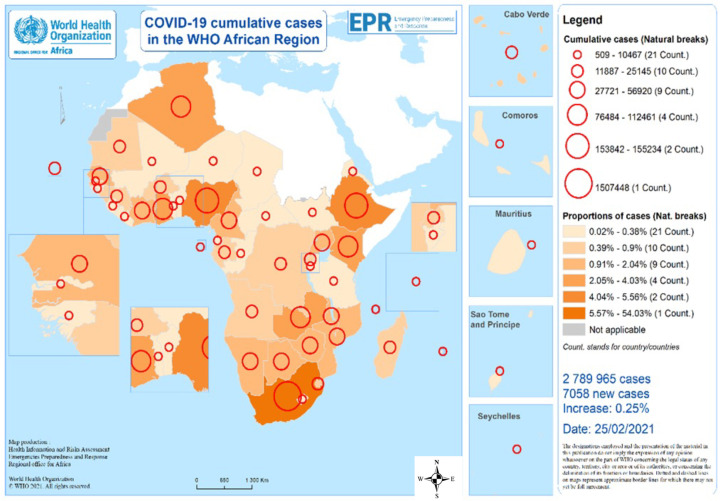
Geographic Distribution of Cumulative COVID-19 in the WHO African Region (Source: IMST presentations).

**Figure 2 tropicalmed-07-00183-f002:**
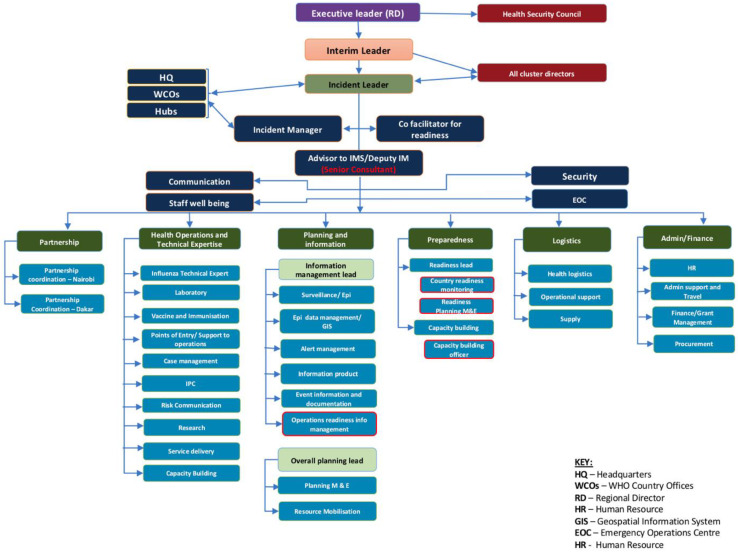
AFRO COVID-19 AFRO Incident Management Support Team (IMST) for preparedness and response (Note: four sub pillar teams have been highlighted by a red blank because their functions were often cross-cutting across the other pillars but also supported countries work).

**Figure 3 tropicalmed-07-00183-f003:**
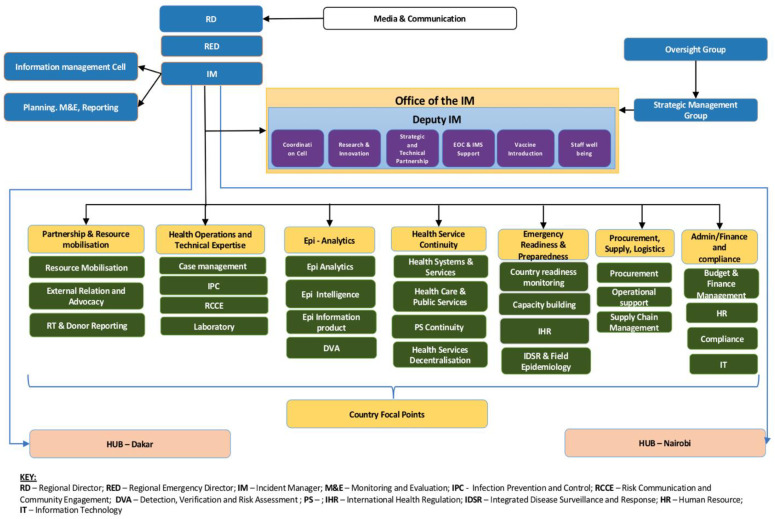
Revised AFRO IMST structure.

**Table 1 tropicalmed-07-00183-t001:** Interventions and activities of HOTE pillar lead, sub-pillars, and cross pillar coordination (from 25 February to 25 February 2021).

Pillar/Sub-Pillar	Interventions	Activities
Pillar Lead (1)	- Provide guidance and leadership of the pillar.	- Coordinate the operations of the five sub-pillars - Represents the pillar in the cross-pillar meetings at the IMST strategic meeting - The lead represents the voice of the pillar and communicates all their needs to the management.
IPC (7)	- Strengthen patients’ treatment and prevent transmission to staff, all patients/visitors, and the community against COVID-19 infection by reviewing, updating, and disseminating existing and interim IPC protocols, including triage.	- Build capacity of health care workers on IPC for COVID-19 and SARIs (staff, training, supplies-PPEs, and equipment) for member states. - Provide strategic guidance to countries on all aspects of the pandemic response relating to the IPC component - Provide countries with the technical recommendations and tools necessary for their application - Strengthen the capacities of countries in the implementation of interventions, as well as in monitoring and evaluation - Strengthen the activities carried out by the other sub-pillars because of their transversal nature
RCCE (6)	- Strengthen public awareness through an integrated risk communication and community engagement approach on the COVID-19, including a psycho-social component in 47 Member States.	- Strengthen the identification of RCCE actions towards specific population groups and settings to address knowledge, rumours, and misinformation in 47 Member states.
CM (5)	- Improve clinical care for COVID-19 patients through slowing and stopping transmission, finding, isolating, and testing every suspected case, and provide timely, appropriate care to affected patients.	- Support clinical CM for COVID-19 patients in Member States’ treatment facilities through training, developing guidance and SOPs, assessments for screening/isolation capacity, ICU units, and related medical supplies access.
Laboratory (5)	- Strengthen and maintain regional and country surveillance systems to gather data on alerts, suspected cases and confirmed COVID-19 cases in collaboration with partners.	- Provide laboratory support at National and Sub-national levels, including reagents and other supplies to the Member States.
EMT (2)	- Strengthen and establish the regional training centre; and the national EMTs	- Enhance collaboration/coordination with Member states, Africa CDC, Regional Economic Communities, National and International NGOs and UN resident coordinators (RCs) to mobilise experts and safe deployment to support the response.

**Note:** () shows the number of staff.

## Data Availability

All required data is present in the manuscript.

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
