# Peer review of "Coordination and Management of COVID-19 in Africa through Health Operations and Technical Expertise Pillar: A Case Study from WHO AFRO One Year into Response"

_tropicalmed, 2022, doi:10.3390/tropicalmed7080183_

Round 1

Reviewer 1 Report

In this study, the authors describes the work done under the health operations and technical expertise (HOTE) pillar of the health operations and technical expertise (IMST) in ensuring effective COVID-19 response. 
In addition, they presented a comprehensive study and proposed recommendations to strengthen the approach.

Author Response

The authors are grateful to the reviewer

Reviewer 2 Report

Review of  tropicalmed (1554618)  Coordination and Management of COVID-19 in  Africa Through Health Operations and Technical Expertise Pillar: A Case Study from WHO AFRO One into Response

Introduction

  • Didn’t see reference to Figure 1 in the text (I apologize if I missed it)

  • The purpose of the study is not clear. In the abstract it is stated This paper documents the lessons learned from the activities of the HOTE pillar and recommends future actions for improving the coordination On page 3 it is stated This paper describes the work done under the HOTE pillar of the IMST in ensuring effective COVID-19 response. These two statements could viewed as different – please revise so they are either the same or better complement each other. 

  • Please explain how the HOTE pillar plays a significant role (or not significant role) in linking WHO with the Ministry of Health (MOH) and partners to ensure optimal coverage and quality of health services in response to emergencies by promoting the implementation of the most effective, context-specific public health interventions and clinical services by operational partners. Is there an example that would document this statement? 

Methods

  • The first statement: The paper used….   The paper didn’t use anything.   Please change to something like The researchers employed a … or  A descriptive, case-study methodology was used in this study.     Even though mixed methods is mentioned later, suggest also including mixed methods in the description – for example, A mixed methods, case-study methodology was used to conduct this this study…..

  • The text highlights that the parameters included all activities implemented by the HOTE.  How is activity defined?  Is an email an activity?  Is a personal communication an activity? These activities are central to this study and need to be defined.  

  • It is not clear how the available reports were collected. What specific actions were used to procure the information needed?  If a researcher wanted to replicate this study, for this part of the study how would one procure this important information? Also, in reviewing the reports, etc.  what specific data was being collected? Was a rubric or data collection template used to consolidate data?

  • More information is needed for the key informant interviews. Was a script developed for the interviews?  Who conducted the interviews and how were they trained?   How was the qualitative data analyzed?  Was informed consent reviewed?   How many total key informants were identified and how many of those contacted declined to participate to get the 14 in this study?

Key Lessons Learned and Challenges

  • The challenges seem to be clearly identified but the lessons learned are always clear. Sometimes the narrative is a summary of what was found or the implication of the finding – the lesson learned is not clear.  To the extent possible try to present the actual lessons learned.

  • The narrative uses the term observations quite a number of times and it’s clear that observation is the right term to use – at least not based on the methodology. For example, the first paragraph on page 11, suggest changing Our observation showed multiple….  consider changing to We found multiple information sharing channels….     Were there actual observations or was this found after reviewing the documents, interviews, etc..    If actual observations were made then the methodology needs to be clear that observations were made. 

  • The limitations of the overall study are not clear.

Author Response

Introduction

  • Didn’t see reference to Figure 1 in the text (I apologize if I missed it)

Thank you for pointing this out. We have now addressed it in line 58 and included the statement: distributed as per Figure 1

  • The purpose of the study is not clear. In the abstract it is stated This paper documents the lessons learned from the activities of the HOTE pillar and recommends future actions for improving the coordination On page 3 it is stated This paper describes the work done under the HOTE pillar of the IMST in ensuring effective COVID-19 response. These two statements could viewed as different – please revise so they are either the same or better complement each other. 

We have now aligned the purpose of the study both in the abstract and the paper as read in lines 23-25 (abstract) and 106-109 (main paper).

  • Please explain how the HOTE pillar plays a significant role (or not significant role) in linking WHO with the Ministry of Health (MOH) and partners to ensure optimal coverage and quality of health services in response to emergencies by promoting the implementation of the most effective, context-specific public health interventions and clinical services by operational partners. Is there an example that would document this statement? 

We have now captured and expounded on how HOTE pillar plays a significant role (or not significant role) in linking WHO with the Ministry of Health (MOH) and partners as suggested by the reviewer in lines 124-132

Methods

  • The first statement: The paper used….   The paper didn’t use anything.   Please change to something like The researchers employed a … or  A descriptive, case-study methodology was used in this study.     Even though mixed methods is mentioned later, suggest also including mixed methods in the description – for example, A mixed methods, case-study methodology was used to conduct this this study…..

We have now edited this section in line 136

  • The text highlights that the parameters included all activities implemented by the HOTE.  How is activity defined?  Is an email an activity?  Is a personal communication an activity? These activities are central to this study and need to be defined.

We have now highlighted that the activities are defined in the comprehensive Strategic Preparedness and Response Plan (SPRP) February-December 2020 updated (May 2020) in lines 140

  • It is not clear how the available reports were collected. What specific actions were used to procure the information needed?  If a researcher wanted to replicate this study, for this part of the study how would one procure this important information? Also, in reviewing the reports, etc.  what specific data was being collected? Was a rubric or data collection template used to consolidate data?

We have now expounded the document review section in lines 136-147 as suggested by the reviewer.

  • More information is needed for the key informant interviews. Was a script developed for the interviews?  Who conducted the interviews and how were they trained?   How was the qualitative data analyzed?  Was informed consent reviewed?   How many total key informants were identified and how many of those contacted declined to participate to get the 14 in this study?

We have now expounded the key informant interviews section in lines 136-188 as suggested by the reviewer.

Key Lessons Learned and Challenges

  • The challenges seem to be clearly identified but the lessons learned are always clear. Sometimes the narrative is a summary of what was found or the implication of the finding – the lesson learned is not clear.  To the extent possible try to present the actual lessons learned.

In this study we tailored the learnings as both positive and negative rather than use a framework of analysis on lessons learnt as also guided by the second reviewer. We believed that using both was essential providing a background for formulating change moving forward driven by the thematic analysis/ content.

  • The narrative uses the term observations quite a number of times and it’s clear that observation is the right term to use – at least not based on the methodology. For example, the first paragraph on page 11, suggest changing Our observation showed multiple….  consider changing to We found multiple information sharing channels….     Were there actual observations or was this found after reviewing the documents, interviews, etc..    If actual observations were made then the methodology needs to be clear that observations were made. 

Throughout the document, we have now replaced the word observations with findings and results to avoid the confusion

  • The limitations of the overall study are not clear.

We have now included this in lines 469-474

Reviewer 3 Report

I appreciate the opportunity to review this interesting work by Ngoy et al.
Although it has been tried to handle it as a case study, as it says in its own text, this manuscript is of the lessons from the field type. This change does not affect the merit of the manuscript, which is relevant.

ABSTRACT
I believe it should be unstructured, with a greater focus on lessons learned.

INTRODUCTION
Update the information on the affected population as of the revision date.
I suggest considering this reference: Tadj A, Lahbib-Seddiki SM. Our Overall Current Knowledge of Covid 19: An Overview Microb Infect Chemother. 2021; 1: e1262. https://doi.org/10.54034/mic.e1262

METHODS
As already mentioned, the merit of this manuscript is high, so it does not need to be treated as an original article.

Author Response

REVIEWER 3

Comments and Suggestions for Authors

I appreciate the opportunity to review this interesting work by Ngoy et al.
Although it has been tried to handle it as a case study, as it says in its own text, this manuscript is of the lessons from the field type. This change does not affect the merit of the manuscript, which is relevant.

We thank the reviewer for this positive comment.

ABSTRACT
I believe it should be unstructured, with a greater focus on lessons learned.

Using this comment and based reviewer 1’s comments, we have reshaped the content of the finding to read ‘key learnings’ in an unstructured way

INTRODUCTION
Update the information on the affected population as of the revision date.

We believe that given that the paper was written and designed and written as of March 2021, the content of the paper with a link to cases and deaths recorded in Africa were analysed as of then. The dynamics of the pandemic has significantly changed and thus using the current statistics would bias the presentation.

I suggest considering this reference: Tadj A, Lahbib-Seddiki SM. Our Overall Current Knowledge of Covid 19: An Overview Microb Infect Chemother. 2021; 1: e1262. https://doi.org/10.54034/mic.e1262

We have read through the paper provided by the reviewer and noted that it only describes what COVID-19 is and which does not fit into our description of content on coordination under HOTE; thus, we have elected not to use it.

METHODS
As already mentioned, the merit of this manuscript is high, so it does not need to be treated as an original article.

We thank the reviewer for this positive comment.

Reviewer 4 Report

- The title is too long, maybe I can shorten it.
- This article cannot be considered a case report, it does not deal with information on symptoms, signs, diagnosis or treatment. It could be included as a guideline if accepted.
- It should describe the parameters studied and all the activities implemented by the HOTE pillar, with their categories. It should describe which variables are studied in the interviews.
- Who is selected for the interview along with the eligibility criteria, number of individuals and type of sampling for selection.
- Describe all statistical methods, including those used to control for confounding and methods used to examine subgroups and interactions.
Results
- Does not explain the criteria used to make the modifications to the HOTE pillar, following its revisions.
- No reference to the interviews conducted throughout the document.
- The paper is really a working paper, well put together as a protocol and as lessons learned that will improve decision making, but it is not the type of paper to be published in this journal.

Author Response

Comments and Suggestions for Authors

- The title is too long, maybe I can shorten it.

To capture all the important elements in the title, we have included the type of study, Where done, and by who? We however, note that shortening Health Operations and Technical Expertise to HOTE may confuse the readers in this instance thus we have elected to leave it as it is. We note that it remains within the word limits.

- This article cannot be considered a case report, it does not deal with information on symptoms, signs, diagnosis or treatment. It could be included as a guideline if accepted.

This article is not a case report per se but rather a research article that used a case study methodology. A case study methodology as described by Yin 2020, is a methodology that focuses on evaluating a phenomenon/ context for which the researcher has no control over. In our case, we had no control over the coordination and implementation of the coordination and management of COVID-19 in Africa. For sure, it is not a guideline as suggested by this reviewer.

 - It should describe the parameters studied and all the activities implemented by the HOTE pillar, with their categories. It should describe which variables are studied in the interviews.

We have now highlighted the parameters studied and the activities as defined in the comprehensive Strategic Preparedness and Response Plan (SPRP) February-December 2020 updated (May 2020) in lines 140. Additionally, table 1 captures the activities in detail. The variables studied in the interviews have now been captured in lines 156-164.

- Who is selected for the interview along with the eligibility criteria, number of individuals and type of sampling for selection.

We have now expounded the key informant interviews section in lines 137-189 as suggested by the reviewer.

- Describe all statistical methods, including those used to control for confounding and methods used to examine subgroups and interactions.

Our study is not a quantitative study that requires control for confounding and methods used to examine subgroups and interactions but rather a qualitative study that has now been adequately described in the methods section

Results

- Does not explain the criteria used to make the modifications to the HOTE pillar, following its revisions.

This has now been provided in lines 215-219

- No reference to the interviews conducted throughout the document.

We have now elaborated this in the method section

 - The paper is really a working paper, well put together as a protocol and as lessons learned that will improve decision making, but it is not the type of paper to be published in this journal.

This is not a working paper, or a protocol as suggested by the researcher but a research article that is qualitative in nature.

Round 2

Reviewer 1 Report

-

Reviewer 2 Report

The revisions are fine.  I recommend publication of the manuscript.

Reviewer 3 Report

Although the improvement of the manuscript is substantial, it does not correspond to an original article.
Authors should observe the instructions to authors and determine the most suitable type of manuscript.

Reviewer 4 Report

Dear authors 
Despite the modifications made, I am still of the opinion that this is an article with very low evidence and that it should not be published in this journal.
Kind regards